# Mucosal Healing in Celiac Disease: Villous Architecture and Immunohistochemical Features in Children on a Long-Term Gluten Free Diet

**DOI:** 10.3390/nu14183696

**Published:** 2022-09-07

**Authors:** Roberta Mandile, Mariantonia Maglio, Caterina Mosca, Antonella Marano, Valentina Discepolo, Riccardo Troncone, Renata Auricchio

**Affiliations:** 1Department of Translational Medical Science, Pediatrics Section University Federico II, via Pansini 5, 80131 Naples, Italy; 2European Laboratory for the Investigation of Food Induced Disease (ELFID), University Federico II, via Pansini 5, 80131 Naples, Italy

**Keywords:** mucosal healing, gluten-free diet, immunohistochemistry, villous architecture

## Abstract

Considerable heterogeneity exists across studies assessing intestinal mucosal recovery in celiac (CD) patients on a gluten-free diet (GFD). We aimed at investigating histological and immunohistochemical features in CD patients on a long-term GFD and to correlate them to the GFD duration. Morphometrical and immunohistochemical analysis were retrospectively performed on duodenal biopsies in three groups of children: 33 on a long-term (>2 years) GFD (GFD-group), four of which remained seropositive despite dietary adherence, 31 with villous atrophy (ACD-group) and 76 heathy, non-celiac (CTR-group). Moreover, in the GFD-group, we correlated immunohistochemical alterations to the GFD duration. The villous to crypt (V/C) ratio significantly improved after the GFD and completely normalized in all patients, becoming even higher than in the CTR-group (median value 3.2 vs. 3, *p* = 0.007). In parallel, the number of CD3+ and TCRγδ+ cells in the epithelium were significantly reduced in the GFD compared to ACD patients, even if they remained higher than in the CTR-group (*p* < 0.05). In contrast, CD25+ cells in the lamina propria significantly decreased after the GFD (*p* < 0.05) and become comparable to the CTR-group (*p* = 0.9). In the GFD-group there was no difference in the immunohistochemical parameters between seropositive and seronegative patients and alterations did not correlate to GFD length. In conclusion, a GFD is able to both restore a normal V/C ratio and reduce inflammation, but the epithelium maintains some stigmata of the disorder, such as an increased number of CD3+ and TCRγδ+ cells. These alterations persist regardless of the duration of the GFD.

## 1. Introduction

Celiac disease (CD) is an immune-mediated systemic disorder elicited by gluten and related prolamins in genetically susceptible individuals [1]. Its incidence is progressively increasing in adult and pediatric populations, due to both an improvement in diagnostic accuracy and a real increase in the incidence [2]. Nowadays, the only available treatment is represented by a strict adherence to a lifelong gluten free diet (GFD), which is the only way to prevent short- and long-term complications of untreated CD [3]. In fact, severe mucosal damage with villous atrophy correlates with the development of future complications and complete mucosal recovery is essential to guarantee a good prognosis [4].

There is considerable heterogeneity across studies assessing the completeness of mucosal recovery achieved by a GFD in CD [5,6,7,8,9]. Some celiac patients are reported to fail to achieve complete mucosal recovery even on a strict dietary regimen, despite this issue seeming to be much more relevant in adult than in pediatric populations [10]. To this regard it must be noted that the European Society for Paediatric Gastroenterology Hepatology and Nutrition (ESPGHAN) recently recommended against routinely repeating biopsies in children on a GFD [11]. Moreover, the “definition” of mucosal recovery remains hard to establish. Historically, mucosal damage was defined as an alteration of normal intestinal architecture, with shortened or even absent villi, hyperplastic crypts, and an increased number of intraepithelial lymphocytes (IELs) upon histological evaluation [12]. Now, more sophisticated techniques, such as flow cytometry and immunohistochemistry have allowed a more extensive evaluation of mucosal damage in CD through the characterization of single types of cells infiltrating both the epithelium and lamina propria [13]. Indeed, a damaged intestinal mucosa in CD is characterized in the epithelium by an increase in ILEs expressing the CD3 surface marker, with around 10% of them also expressing the γδ receptor, and in the lamina propria by an increase in inflammatory mononuclear cells that express the CD25 surface marker [14]. These cells tend to decrease on a GFD, but many studies suggest that some stigmata of the disorder remain despite clinical and serological remission [15,16,17,18]. In addition, it remains to be established whether these subtle alterations may influence the patient’s prognosis.

In the present study, we aim to investigate histological and immunohistochemical features in CD patients on a long-term GFD and to correlate them to the duration of the GFD.

## 2. Methods

### 2.1. Patients and Study Design

This retrospective study was carried out at the University Federico II of Naples, a tertiary care centre and the reference centre for the diagnosis and management of CD in Campania, Italy. The study group (GFD group) included children with a diagnosis of CD on a long-term GFD (inclusion criteria: more than 2 years, mean duration time 8 years, range 1.9–18 years). The adherence to a correct dietary regimen was assessed both by CD specific serology and by an expert nutritionist performing a detailed nutritional interview. We selected two control groups: CD patients with villous atrophy at diagnosis (ACD group), and healthy subjects with normal small intestinal mucosa architecture (CTR group). This latter group included subjects with negative CD serology and a diagnosis of one of the following clinical conditions: eosinophilic esophagitis, type 1 diabetes, functional gastrointestinal disorders, and first-degree relatives of CD patients. Patients with inflammatory bowel diseases or other inflammatory conditions of the lower gastro-intestinal tract were excluded. Patients received a duodenal biopsy during an esophagogastroduodenoscopy (EGDS) performed for diagnostic purposes, after giving their informed consent. The demographic and clinical features of the patients are summarized in Table 1.

### 2.2. Histological Evaluation

In each patient, EGDS with 5 biopsies, 1 from the bulb and 4 from the distal duodenum, was carried out. According to our protocol, 4 of 5 fragments, including 1 from the bulb, were fixed in 10% formalin, embedded in paraffin, sectioned at a 5 µm thickness and then stained with hematoxylin/eosin. Histological and morphometrical analyses with measurement of villi and crypts by light microscopy were performed by two experienced pathologists. A villous height:crypt depth ratio equal or higher than 2 was considered normal, as stated in latest ESPGHAN guidelines [1]. Among biopsies with a normal villous height:crypt depth ratio, Marsh 0 was defined by the presence of less than 25 intraepithelial lymphocytes (IELs) per 100 enterocytes, and Marsh 1 by the presence of more than 25 IELs per 100 enterocytes. The Marsh score was given based on the score of the fragment with the worst picture. The pathologists were blinded to the serology results.

### 2.3. Immunohistochemical Evaluation

One fragment (not from the bulb) was added to an optimal cutting temperature compound (Killik; BioOptica, Milan, Italy), stored at −80 °C, and used for immunohistochemical staining for CD3+, TCRγδ+, and CD25+ cells, as previously reported [19]. The number of stained cells per millimeter of epithelium determined the density of the cells expressing CD3 and TCRγδ in the intraepithelial compartment. The cut-off values for CD3+ and TCRγδ+ cells were 34 mm and 3.4 mm per epithelium, respectively. On the other hand, the number of cells expressing CD25 in the lamina propria was evaluated within a total area of 1 mm^2^. The cut-off value for CD25+ cells was 4 per mm^2^ of the lamina propria. To determine the cut-off values, 100 children with untreated celiac disease and 50 non–celiac disease control children were studied. The percentiles were obtained using the Statistical Package for the Social Sciences SPSS software (IBM, Chicago, IL, USA). The cut-off values represented the 90th percentile of control patients [19].

### 2.4. Statistical Analysis

All variables were chosen before the analyses because of their possible relevance for the study aims. Quantitative data were expressed using means with 95% confidence intervals. When comparing means, an independent-sample *t* test with 2-tailed significance was used in normally distributed variables, and the Mann-Whitney U test with nonparametric variables. The Chi-square χ^2^ test or Fisher’s exact test were used to test differences between categorical parametric and non-parametric variables, respectively. To test correlation, the Spearman rank-order was used. 

## 3. Results

### 3.1. Patients

The study group included 38 CD subjects in clinical remission and already on a strict GFD (mean duration time 8 years, range 1.9–18 years), as assessed by nutritional interview performed by an expert dietician. The single most frequent reason (in 33/38 patients) to obtain a biopsy from patients on the GFD was the need to have a pre-challenge biopsy in the context of clinical studies. After the gluten challenge, all patients involved in the study eventually relapsed. These patients had a negative CD-associated serology and did not complain of any symptoms. Only 4/38 patients repeated a biopsy because of the persistence of a positive CD serology (EMA and/or anti-TG2) despite the nutritional interview assessing good adherence to the correct dietary regimen and these patients also being completely asymptomatic. In 1/38 patient, an upper endoscopy with biopsies was performed because of suspected gastroesophageal reflux disease. For the control groups, we selected both patients with a negative CD associated serology (N = 44) and CD patients at diagnosis (N = 31). There were no significant differences between the three groups in the sex distribution or other demographic features (Table 1).

### 3.2. Histological and Immunohistochemical Evaluations

In total, 23/38 duodenal samples from the GFD group were defined as a Marsh0 at the histological evaluation, while 15/38 were Marsh1. No villous atrophy was found in our cohort of patients, and the V/C ratio was always higher than 2 (Table 2).

The V/C ratio significantly improved after the GFD and completely normalized in all patients, becoming even higher than in the CTR group (median value 3.2 vs. 3, *p* < 0.01) (Figure 1A) though measurement of villus height and crypt depth did not show appreciable differences between the first (mean ± SEM: 333.5 ± 11 and 99.7 ± 3 µm, respectively) and the second (336.2 ± 7.5 and 107.7 ± 3.5 µm) group (Figure 1B). In parallel, densities of CD3+ and TCRγδ+ cells in the epithelium were significantly reduced in the GFD (37 ± 2.8 and 9.8 ± 1.1 cells/mm of epithelium, respectively) compared to the ACD group (66.8 ± 7 and 20.32 ± 2.9 cells; *p* < 0.001 and *p* < 0.01, respectively), even if it remained higher than in the CTR group (26.3 ± 1.8 and 2.6 ± 0.4 cells, respectively; *p* < 0.01 and *p* < 0.0001 respectively) (Figure 2A,B).

In the study group there was not only a higher density of intraepithelial lymphocytes (IELs) but also a higher percentage of subjects with an increase in CD3+ (15/27, 56%; Figure 3A) and TCRγδ+ (23/27, 85%, Figure 3B).

Regarding the IELs compared to the CTR group (8/44, 18%, *p* < 0.0.01; 13/44, 29%, *p* < 0.0001, respectively), of note is that the density of the TCRγδ+ IELs remained altered even after years of a gluten free diet (Figure 2B). In contrast, the number of CD25+ cells in the lamina propria were significantly reduced after following a GFD (8.9 ± 2.4 cells/mm^2^ of lamina propria) and became comparable to the CTR group (8.6 ± 1.6) (*p* = 0.9) (Figure 2C). The percentage of GFD patients showing increased CD25+ cells in the lamina propria was also comparable to the CTR subjects, being 52% (14/27) in GFD vs. 47.7% (21/44) in the CTR, (Figure 3C). In the GFD group, 7/27 (26%) patients showed all three immunohistochemical parameters (CD3+, TCRγδ+ IELs, CD25+) altered at the same time. Interestingly, no differences were observed in the immunohistochemical parameters between seropositive GFD and seronegative GFD patients (Figure 2). Furthermore, in seropositive children, the V/C ratio was always higher than 2, half of them (50%) showed an increased number of CD3+ and CD25+ cells, 75% had an increased number of TCRγδ+ IELs. No correlation was shown between the densities of the cells (CD3+, TCRγδ+ and CD25+) and GFD duration (Figure 4).

## 4. Discussion

In CD, gluten activates an adaptive immune response that ultimately causes damage to the small intestinal mucosa and its exclusion from the diet is currently accepted as the only effective treatment. Different studies carried out in adults and in children, suggest however, that diet is not always able to restore the normal intestinal architecture [4,5,6,7,8,9,10]. Moreover, even when it happens, more subtle signs of mucosal inflammation can persist. Our data, retrospectively obtained from a cohort of celiac children on a long-term GFD, demonstrated that a GFD is effective in restoring a normal intestinal architecture in children, even in those who underwent a second duodenal biopsy for a persistently positive serology despite good adherence to the dietary regimen. Indeed, morphometrical analysis revealed that, in patients on a GFD, the V/C ratio became even higher than in controls. This apparently surprising result can be explained by the fact that enterocyte proliferation in crypts is notably increased in CD patients in remission [20], whose cells are committed to “repairing” damaged tissue, compared to controls. Our results are in contrast with those recently published by Leonard et al. in a retrospective study on 103 CD patients from two children’s hospitals in Boston, who showed that twenty percent of children with CD do not heal on a GFD [21]. In all our patients, the V/C ratio reverted to a normal value after a GFD was commenced. Vice versa, in line with the ESPGHAN [11], our data support the notion that there is no need to repeat biopsies as routine practice in CD children in remission, with a positive impact on patients’ comfort, quality of life and on medical costs.

In contrast with the morphometrical analysis, our immunohistochemical analysis revealed that, despite following a long-term GFD, some signs of inflammation persist in the epithelium of CD patients, while disappearing from the lamina propria [22]. Furthermore, we demonstrated that IELs are significantly reduced in patients on a GFD compared to CD patients at diagnosis but remain significantly increased compared to controls. This is particularly evident for the TCRγδ+ IELs, that remain altered in 85% of patients on a long-term GFD, while lamina propria CD25+ cells completely normalize on a GFD and become comparable to controls.

It is obvious to expect that GFD causes a reduction in the IELs number; however, the literature is not clear whether they completely return to normal values. IELs represent a large population of antigen-experienced “innate-like” T cells that are typically recalled in the intestinal epithelium by inflammatory cytokines released by gluten specific CD4+ T cells activated by gluten in the lamina propria [15,22]. Most IELs express an αβ-receptor and their classical function is to kill stressed epithelial cells in an antigen independent manner. A small proportion of IELs, around 10% of the total, bear the γδ-receptor: the precise functional role of these cells is not fully understood, but recent studies suggest they could have a cytolytic phenotype, expressing high levels of granzyme B under basal conditions and undergoing a reshaping of their function when tissue damage occurs, acquiring a proinflammatory INF γ-producing phenotype in overt CD [23]. Furthermore, the authors revealed, via transcriptional studies, a permanently altered program in TCRγδ+ cells of CD patients irrespectively of a strict adherence to a GFD, suggesting that TCRγδ+ cells represent a hallmark of CD that persist even upon gluten withdrawal from the diet. Besides, it has been previously demonstrated by our and other groups that interleukin (IL)15, that has a prominent role in the recruitment of TCRγδ+ IELs [24], is persistently elevated in patients on a GFD [25,26,27] and could thus be partially responsible for the persistence of TCRγδ+ cells in the intestinal epithelium.

The concept that some inflammatory features persist in CD patients in remission is not completely new in the literature. In a recently performed clinical trial on 19 well-treated adult celiac patients, proteomic analysis of total tissue or the isolated epithelial cell compartment from intestinal biopsies collected before and after a 14-day gluten challenge, demonstrated that patients with a stronger mucosal response to the challenge already displayed signs of ongoing tissue inflammation before the gluten challenge [28]. This minimal tissue inflammation in basal conditions is paralleled by increased gluten specific CD4+ T-cell frequencies in the gut and the presence of a low-level blood inflammatory profile. Thus, apparently, in well-treated subjects, the disease cannot be completely quiescent, with the presence of low-grade inflammation and anti-gluten immunity in the gut mucosa and histological evaluation not necessarily correlated with a full recovery [28]. In addition, from a genomic point of view, Dotsenko et al. recently published that 167 genes were differentially expressed in the intestinal mucosa of CD patients on a GFD and after a gluten challenge. In particular, genes encoding proteins that transport small molecules were expressed less, suggesting that GFD patients were not completely “healed” despite histological remission [29].

Our data also suggest that mucosal healing is independent of both the duration of the diet and the persistence of a positive CD associated serology in patients that correctly follow the dietary regimen. We noticed, in fact, that the number of CD25+ cells and IELs did not correlate to the years of following the GFD. Since the biopsy in remission was performed after at least 2 years on a GFD (mean time 8 years, range 2–18 years), we could speculate that, after that period of time, the intestinal mucosa acquires its definitive shape. Moreover, in the small subset of children that repeated the intestinal biopsy because of the persistence of a positive CD associated serology despite a good adherence to the GFD (as assessed by the nutritional interview), the immunohistochemical features were comparable to the whole cohort. The presence of circulating anti-transglutaminase antibodies could be due to persistent intestinal antibody production by specific plasma cells even years after gluten withdrawal and is not necessarily indicative of insufficient dietary compliance [30,31].

In conclusion, our work supports the idea that a GFD is an effective strategy both to restore normal intestinal architecture and to reduce inflammation in the lamina propria of CD children; however, the epithelium maintains some stigmata of the disorder. The increased number of TCRγδ+ cells despite a long-term GFD enforces the concept that these cells represent a hallmark of the disease, and this compartment remains altered regardless of gluten consumption. Moreover, the mucosal reshaping associated with tissue healing takes place mostly in the first 2 years of a GFD and does not necessarily correlate with the presence of a positive CD serology. Further studies are needed to address the issue of whether these subtle immunohistochemical alterations, despite the restoration of a normal intestinal architecture, can have a clinical impact in terms of the development of long-term complications in CD patients.

## Figures and Tables

**Figure 1 nutrients-14-03696-f001:**
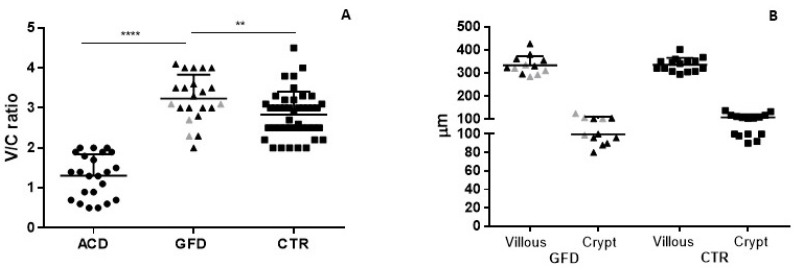
(**A**) Villous/crypts ratio in patients with villous atrophy, on a gluten free diet and in control patients. (**B**) Villi and crypts length in patients on a gluten free diet and in control patients. Grey dots represent patients on a gluten free diet with a persistently positive CD-serology. V/C: villous/crypts ratio; ACD: atrophic celiac disease; GFD: gluten free diet; CTR: controls. *: level of statistical significance; ** means *p* < 0.005, **** means *p* < 0.00005.

**Figure 2 nutrients-14-03696-f002:**
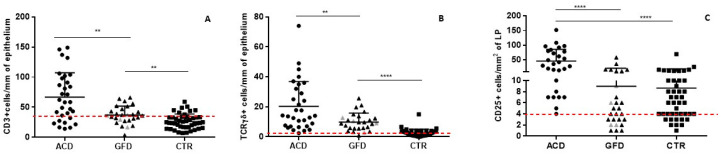
Immunohistochemical parameters in different patients group. (**A**) CD3+ cells/mm of intestinal epithelium. (**B**) TCRγδ+ cells/mm of intestinal epithelium. (**C**) CD25+ cells/mm^2^ of intestinal lamina propria. Grey dots represent patients on a gluten free diet with a persistently positive CD-serology. Dashed horizontal line represents cut-off limits of normality for each immunohistochemical parameter. ACD: atrophic celiac disease; GFD: gluten free diet; CTR: controls. *: level of statistical significance; ** means *p* < 0.005, **** *p* < 0.00005.

**Figure 3 nutrients-14-03696-f003:**
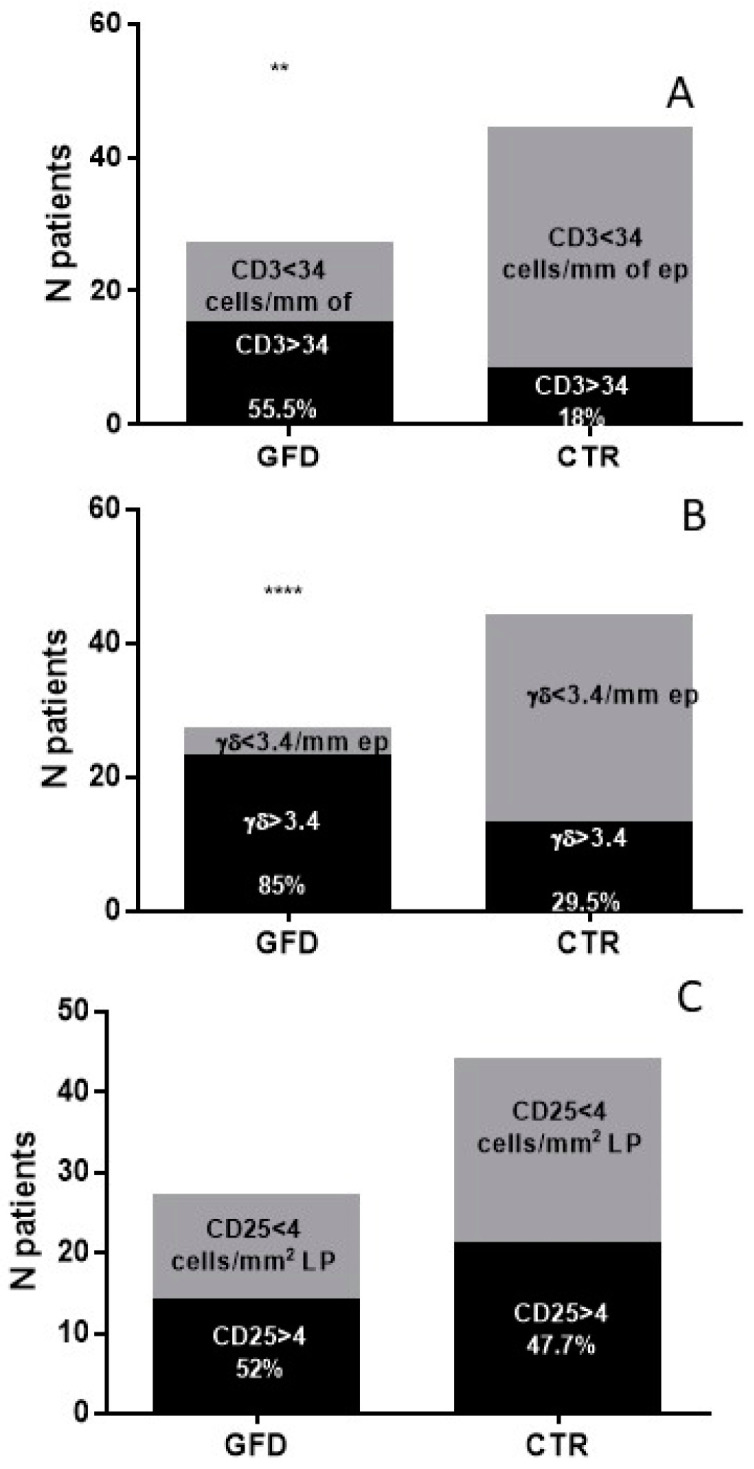
Percentage of patients with altered immunohistochemical parameters. (**A**) patients with altered CD3+ cells/mm of intestinal epithelium. (**B**) patients with altered TCRγδ+ cells/mm of intestinal epithelium. (**C**) patients with altered CD25+ cells/mm^2^ of intestinal lamina propria. N: number; GFD: gluten free diet; CTR: controls. *: level of statistical significance; ** means *p* < 0.005, **** *p* < 0.00005.

**Figure 4 nutrients-14-03696-f004:**
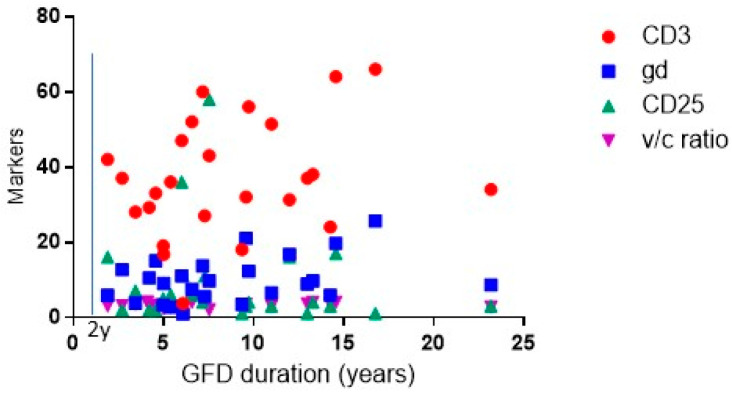
Linear correlation among different immunohistochemical markers and gluten free diet duration. Red circle represents CD3+ cells; blue squares TCRγδ+ cells, green up-pointed triangles CD25+ cells, violet down-pointed triangles villous/crypts ratio. Vertical blue line represents the minimum period of gluten free diet (2 years). GFD: gluten free diet.

**Table 1 nutrients-14-03696-t001:** Clinical features of enrolled patients.

	GFD Group	ACD Group	CTR Group
Number	28	31	43
Sex (female/male)	16/12	20/11	20/23
Age at diagnosis Mean yrs (range)	6.8 (0.5–17.4)	7.5 (2.1–15.5)	6.6 (0.8–17.2)
GFD duration Mean yrs (range)	8.6 (1.9–18)	-	-
Anti-TG2 Mean ULN value	0.67xULN	8.17xULN	0.1xULN

GFD: gluten-free diet; ACD: atrophic celiac disease; CTR: controls; anti-TG2: anti-transglutaminase antibodies; yrs: years; ULN: upper limit of normality.

**Table 2 nutrients-14-03696-t002:** Histological and Immunohistochemical features.

	GFD Group	ACD Group	CTR Group
V/C	3.23	1.3	2.85
CD3+ cells/mm (mean ± SD)	37.0 ± 1.41	66.8 ± 36.99	26.4 ± 13.77
TCRγδ+ cells/mm (mean ± SD)	9.8 ± 0.98	20.3 ± 14.77	2.6 ± 5.59
CD25+ cells/mm^2^ (mean ± SD)	8.9 ± 10.95	45.8± 36.03	7.3 ± 10.9

GFD: gluten-free diet; ACD: atrophic celiac disease; CTR: controls; V/C: villous/crypts ratio; SD: standard deviation.

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
