# Peer review of "Mucosal Healing in Celiac Disease: Villous Architecture and Immunohistochemical Features in Children on a Long-Term Gluten Free Diet"

_nutrients, 2022, doi:10.3390/nu14183696_

Round 1
Reviewer 1 Report
Section 2.1 Please clearly indicate when (how long after applying GFD) the investigated biopsy was performed.
“The study group (GFD group) included children with a diagnosis of CD “ Please correct the sentence as from data in Table 1 it can be deduced that also adults were participants of the study (the longest GFD duration is over 23 years).
Section 3.1 and Table 1. Which data are proper? “on a strict GFD (mean duration time 8 years, range 2-18 years)” or in the Table 1 – the range of GFD duration: “1.9-23.18”
Page 7
“Maki et al” is missing in Reference section. Please specify what tissue was investigated by these authors.
Refence section
Position 22 – shouldn’t it be merged with 21?
Please add "Maki et al" reference
Author Response
We would like to thank the Editorial Board and the reviewers for the overall positive appraisal of the manuscript and the constructive feedback. We amended the manuscript according to the reviewers’ suggestions. We believe the suggested edits allowed to significantly improve the manuscript, making it clearer. We hope the revised manuscript will be suitable for publication in Nutrients. Below, a detailed point by point reply to the reviewer’s comments.
Section 2.1 Please clearly indicate when (how long after applying GFD) the investigated biopsy was performed.
Specified now in the text
“The study group (GFD group) included children with a diagnosis of CD” Please correct the sentence as from data in Table 1 it can be deduced that also adults were participants of the study (the longest GFD duration is over 23 years).
We would like to thank the reviewer for pointing this incongruence out and allowing us to make the appropriate edits. The range of GFD duration was misreported in the table: it was actually 2.3 years and not 23 years. This is now corrected in the Table 1.
In this study we included only patients who were diagnosed in a paediatric age. However, one of them, was re-biopsied at the age of 18, after 18 years of GFD, in fact he was diagnosed in the first year of life. This represented the longest GFD duration.
Section 3.1 and Table 1. Which data are proper? “on a strict GFD (mean duration time 8 years, range 2-18 years)” or in the Table 1 – the range of GFD duration: “1.9-23.18”).
The longest GFD duration was 18 years and it is referred to the above specified individual case, in this specific case, the patient was diagnosed in the first year of life right after weaning.
In line with this, the following is correct “on a strict GFD (mean duration time 8 years, range 2-18 years)”, while in Table 1 the range of GFD duration was edited as follows: “1.9-18”).
Since in Italy 18 years is the threshold for paediatric age, we confirmed that the study was performed on a paediatric cohort.
Page 7, “Maki et al” is missing in Reference section. Please specify what tissue was investigated by these authors.
Thank you again for noticing this missing reference. Indeed, the correct reference to be cited here is Dotsenko et al (corresponding to reference number 29). The study was performed on the intestinal mucosa of CD patients on a GFD and after gluten challenge. We now included the correct reference and specify the studied tissue in the text.
Refence section
Position 22 – shouldn’t it be merged with 21?
Thank you very much for noticing, we merged the two references.
Please add "Maki et al" reference
As specified above, the correct reference to be cited in place of “Maki et al” was reference number 29 by Dotsenko et al., indeed Maki M. was a co-author but not the first author of the manuscript. We now include only reference number 29 and cite that.
Reviewer 2 Report
This paper describes the histological findings in CD patients on a glutenfree diet for at least 2 years, compared to findings in CD patients at diagnosis and compared to normal controls. The results are retrospectively collected.
My main criticism concerns the lack of specification of the reason for rebiopsing the patients on GFD (except for 4/38 in whom serological tests remained abnormal). Please specify the indication for repeating the endoscopy in the patients. Also, no clinical data on the evolution on GFD are given (anthropometry? complaints?). Without this information, it is difficult to ascertain the representativity of the subjects for GFD compliant asymptomatic patients.
I also wonder whether the subject is relevant to the journal (Nutrients), and would not be more suited for a journal focusing on histology or gastroenterology.
The paper needs minor spelling corrections. For example, p. 2 line 4 ...despite this issue seeming to be...and section 3.1, line 5: ...the nutritional interview confirming good adherence
Author Response
Dear Editor and Reviewers,
We would like to thank the Editorial Board and the reviewers for the overall positive appraisal of the manuscript and the constructive feedback. We amended the manuscript according to the reviewers’ suggestions. We believe the suggested edits allowed to significantly improve the manuscript, making it clearer. We hope the revised manuscript will be suitable for publication in Nutrients. Below, a detailed point by point reply to the reviewer’s comments.
This paper describes the histological findings in CD patients on a glutenfree diet for at least 2 years, compared to findings in CD patients at diagnosis and compared to normal controls. The results are retrospectively collected.
My main criticism concerns the lack of specification of the reason for rebiopsing the patients on GFD (except for 4/38 in whom serological tests remained abnormal). Please specify the indication for repeating the endoscopy in the patients. Also, no clinical data on the evolution on GFD are given (anthropometry? complaints?). Without this information, it is difficult to ascertain the representativity of the subjects for GFD compliant asymptomatic patients.
We would like to thank the Reviewer, for this observation that allowed us to expand on the studied patients’ cohort. The single most frequent reason to obtain a biopsy from patients on gluten free diet was the need of having a prechallenge biopsy. After gluten challenge. all patients involved in the study eventually relapsed (33/38 patients). These patients had a negative CD-associated serology and did not complain of any symptoms. As you correctly pointed out, only 4/38 repeated a biopsy because of the persistence of a positive CD serology (EMA and/or anti-TG2) despite a good adherence to the gluten-free dietary regimen and also these patients were completely asymptomatic. In 1/38 patient upper endoscopy with biopsies was performed because of suspected gastroesophageal reflux disease. The adherence to the GFD was assessed through nutritional interview performed by an expert dietician before performing the duodenal biopsy. These details are now specified in the text.
I also wonder whether the subject is relevant to the journal (Nutrients), and would not be more suited for a journal focusing on histology or gastroenterology.
The paper needs minor spelling corrections. For example, p. 2 line 4 ...despite this issue seeming to be...and section 3.1, line 5: ...the nutritional interview confirming good adherence
We would like to thank again the reviewers for this comment that allowed us to correct all minor spelling mistakes throughout the text
Round 2
Reviewer 2 Report
The authors have adequately responded to the remarks in the first review. There are still some minor spelling errors (eg page 7 line 19: ...responsible in for the ...)